# Developing a data repository to support interdisciplinary research into childhood stunting: a UKRI GCRF Action Against Stunting Hub protocol paper

Kaitlin Conway-Moore [1], Darius Tetsa Tata,[2] Peter Wood,[3] Val Katerinchuk,[3] D M Dinesh Yadav [2], Little Flower Augustine,[4] Manne Munikumar,[4] Assana Diop,[5] Fassiatou Tairou,[5] Modou Lamin Jobarteh [2], Bharati Kulkarni,[4] Babacar Faye,[5] Paul Haggarty,[6] Claire Heffernan[2,7]

For numbered affiliations see end of article.

**Correspondence to**
Professor Claire Heffernan;
Claire.Heffernan@lidc.ac.uk

## ABSTRACT

**Introduction** As a topic of inquiry in its own right, data management for interdisciplinary research projects is in its infancy. Key issues include the inability of researchers to effectively query diverse data outputs and to identify potentially important synergies between discipline-specific data. Equally problematic, few semantic ontologies exist to better support data organisation and discovery. Finally, while interdisciplinary research is widely regarded as beneficial to unpacking complex problems, non-researchers such as policy-makers and planners often struggle to use and interrogate the related datasets. To address these issues, the following article details the design and development of the UKRI GCRF Action Against Stunting Hub (AASH)'s All-Hub Data Repository (AHDR).

**Methods and analysis** The AHDR is a single application, single authentication web-based platform comprising a data warehouse to store data from across the AASH's three study countries and to support data querying. Four novel components of the AHDR are described in the following article: (1) a unique data discovery tool; (2) a metadata catalogue that provides researchers with an interface to explore the AASH's data outputs and engage with a new semantic ontology related to child stunting; (3) an interdisciplinary aid to support a directed approach to identifying synergies and interactions between AASH data and (4) a decision support tool that will support non-researchers in engaging with the wider evidence-based outputs of the AASH.

**Ethics and dissemination** Ethical approval for this study was granted by institutional ethics committees in the UK, India, Indonesia and Senegal. Results will be disseminated via publications in peer-reviewed journals; presentations at international conferences and community-level public engagement events; key stakeholder meetings; and in public repositories with appropriate Creative Commons licences allowing for the widest possible use.

## WHAT IS ALREADY KNOWN ON THIS TOPIC
⇒ Interdisciplinary research collaborations allow for the development of innovative solutions to complex problems.
⇒ Data management required for large-scale interdisciplinary research projects can pose a significant challenge to researchers.

## WHAT THIS STUDY ADDS
⇒ Insight into designing a unique data management system for the production of interdisciplinary research.
⇒ Recommendations for developing an integrated structure capable of supporting interdisciplinary data analysis, storage, querying and decision support.

## HOW THIS STUDY MIGHT AFFECT RESEARCH, PRACTICE OR POLICY
⇒ By identifying a range of data management innovations through our own large-scale interdisciplinary research project, this work has the potential to inform the field.

## INTRODUCTION

Interdisciplinary research centres on the creation of new knowledge from the explicit combination of different disciplines, sub-disciplines or topics, with interdisciplinary research and the underpinning collaboration recognised to be better at unpacking complex problems and supporting innovative solutions. Despite growing attention in recent years to the merits of interdisciplinary research, however, tools for effective data management remain limited.[1 2] As a result, interdisciplinary data analysis is often done in an ad hoc manner, with researchers often left to their own devices.[3 4] Relatedly, there is little guidance in the literature on how to manage data from multiple disciplines to identify the most promising synergies and inter-relationships between

such data. To address these gaps, researchers are moving to develop semantic ontologies or agreed vocabularies to search and organise data around particular topics.

While the development of semantic ontologies clearly enhances the ability of researchers to explore and analyse data across multiple disciplines within a particular field, the needs of other stakeholders of these research outputs are less well addressed. In particular, the ability of policy-makers and planners to mine interdisciplinary datasets is limited. For example, there are currently a handful of widely used global datasets related to child nutrition (eg, WHO Global Database on Child Growth and Malnutrition, the World Bank Health Nutrition and Population Statistics, the UNICEF-WHO-World Bank Joint Child Malnutrition Estimates or indeed, the University of Oxford's Young Lives dataset), yet these tools do not explicitly focus on cross-disciplinary synergies within their data outputs.

Seeking to address these issues and inform the emerging field of interdisciplinary data management, the following article details the protocol for developing the UKRI GCRF Action Against Stunting Hub (AASH)'s All-Hub Data Repository (AHDR). The Hub's 18 institutional collaborators will collect and analyse approximately 800 000 data points from study participants in India, Indonesia and Senegal. As such, we are collecting a diverse array of data from surveys, direct observations and anthropometric measurements to focus group discussions and in-depth qualitative interviews to biological and environmental samples.

The AHDR is underpinned by three key questions:
1. What is the best platform for the storage and querying of a diverse three-country comparative dataset related to child stunting?
2. What does a new semantic ontology for child nutritional research look like?
3. Can such a data management tool also be used to create a decision-support tool relevant for diverse audiences of policy-makers and planners working on child stunting globally?

A key aim of the AHDR will be to make anonymised AASH data public facing as part of the UKRI's open research data strategy, which seeks to make research processes and findings as transparent, understandable and reproducible as possible. While access to AASH data will be restricted to project staff during the project and initial data analysis period (18 months from the project end date), at the end of this period, all restrictions on data access will be removed and data licences updated to enable free download. In line with UKRI guidelines, these data will be accessible for 10 years after the project ends, and held on a London School of Hygiene & Tropical Medicine server for this period.

## METHODS AND ANALYSIS
### All-Hub Data Repository Design
#### The Data Warehouse, Data Discovery Tool and Interdisciplinary Aid

The AHDR data warehouse will integrate and store AASH data from the various field-level data collection tools, including CommCare, KoboCollect, REDCap and laboratory analyses using an automated Integration Platform as a Service (iPaaS) middleware system, powered by OpenFn.[5–8] The iPaaS middleware will either link to these data collection platforms directly via API connections, or to Secure File Transfer Protocol servers that will contain ready-to-export data in CSV or XLS formats. To ensure that there is alignment in the types of data collected across our three study countries, all metadata from our data collection applications have undergone an extensive process of labelling, mapping and review.

To prepare data stored in the data warehouse for analysis by end users, measures will be put in place for automated data cleansing both at the point of collection and during the data integration process. Specifically, data validation rules and skip logic will be implemented in different Clinical Data Management Applications (CDMAs) to reduce the risk of erroneous and inconsistent data. In addition to this, external scripts will be written in R, STATA and Python to enforce data quality checks in the event they cannot be implemented into the CDMAs, and for the purposes of anonymising/pseudoanonymising data prior to transfer into the data warehouse. Data integration programmes will also be designed to map data from across the AASH by harmonising common variable names and labels; standardising values; calculating derived variables; and recoding data as required. Data collection and transformation will be done in accordance with AASH study protocols, standard operating procedures, General Data Protection Regulation guidelines and both national and local regulations for data protection in Senegal, India and Indonesia.

The data warehouse has been built using PostgreSQL, an open-source database management system that specialises in securely storing complex datasets.[9] Genomic data collected as part of our study will be held on the European Geno-phenome Archive and connected to the AHDR using APIs. Each of our three project countries will also be equipped with high-performance computing servers that allow for the maintenance, computing and analysis of these large files. Researchers seeking access to genomic data will be able to do so via the AHDR, following completion of consent and authorisation forms, as required. The AHDR's data warehouse will also store a limited set of derived and summary variables related to genomic data.

A Data Discovery Tool, seen in figure 1, is being implemented to enable researchers and other end users to browse and download data from the data warehouse. This functionality will enable both viewing and downloading in raw CSV file format and a CSV download displaying metadata labels in place of encoded column names and multichoice values. To facilitate the cross-disciplinary

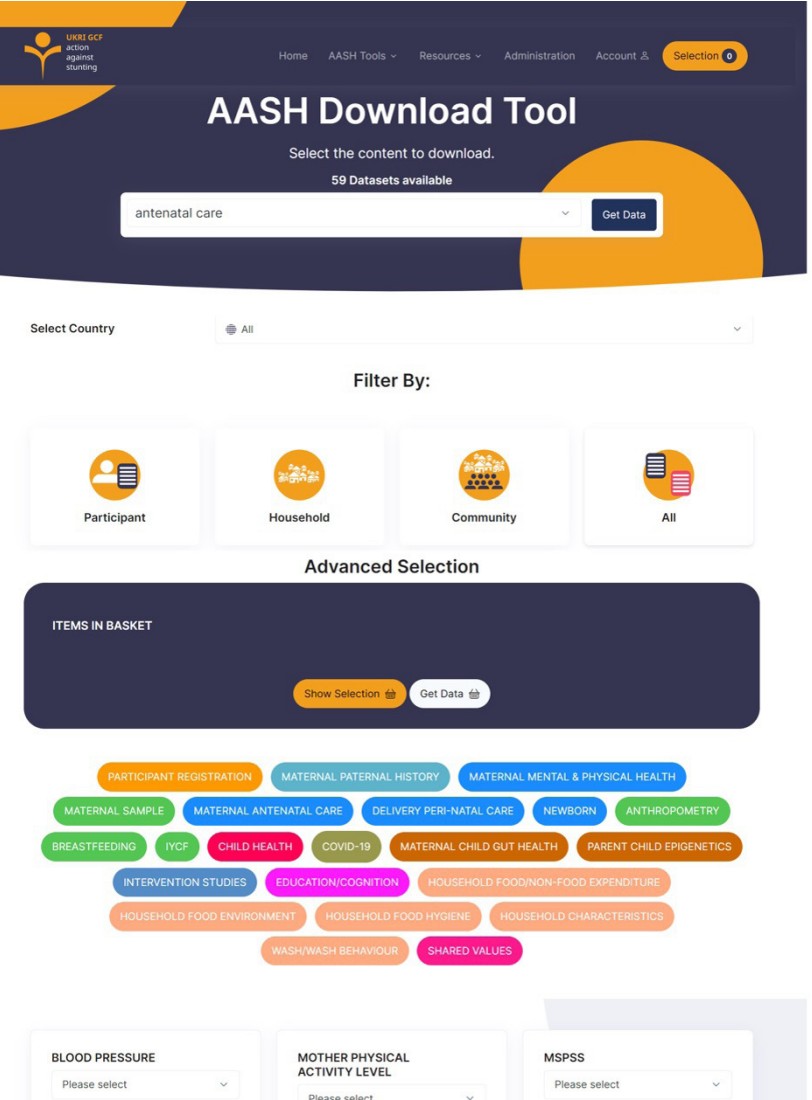

**Figure 1** Data discovery tool for the data warehouse.

analysis, it will be possible to navigate through individual columns of the various data tables and combine them into a single CSV output, joined on a unique subject identification number, or alternative identifiers when a subject identification number is not present.

The Interdisciplinary Aid (IA) will be created by linking the various sources of data collected across our project sites to both topic-based and conceptual entities through a Neo4j graph database system (Version 4). All data variables will be tagged and allocated to a position within a defined hierarchy using specific domain ontologies. The IA will enable the establishment of direct connections between variables based on their syntactic matching with data tags and/or within the hierarchy. The tool will also enable the establishment of indirect connections between variables and datasets based on their semantic proximity (ie, ontological connection) with the aid of semantic matching algorithms. Researchers will then have a visual interface that they can browse to narrow in on the types of data they find most useful to their work,

while also finding conceptual/logical connections to data outside their primary discipline. Finally, research collaboration will also be supported via the platform's use of CKAN (V.2.9), an open-source data portal. With CKAN, researchers will be able to upload analysed and partially analysed datasets for use by other researchers both within and outside of their discipline. In doing so, the building blocks of interdisciplinary analysis will be made available to all end users.

### Metadata Catalogue and Semantic Ontology
The AHDR metadata catalogue will provide an overview of all the AASH data available for download and analysis. Underpinning the metadata catalogue will be a new semantic ontology for child stunting. To create the semantic ontology, first, terms from existing ontologies from across the thematic disciplines involved in the AASH (eg, MeSH, FoodOn, ECCD Ontology) will be mined.[10 11] Second, expert consultations will take place to check, define and agree terms. The terms will then be

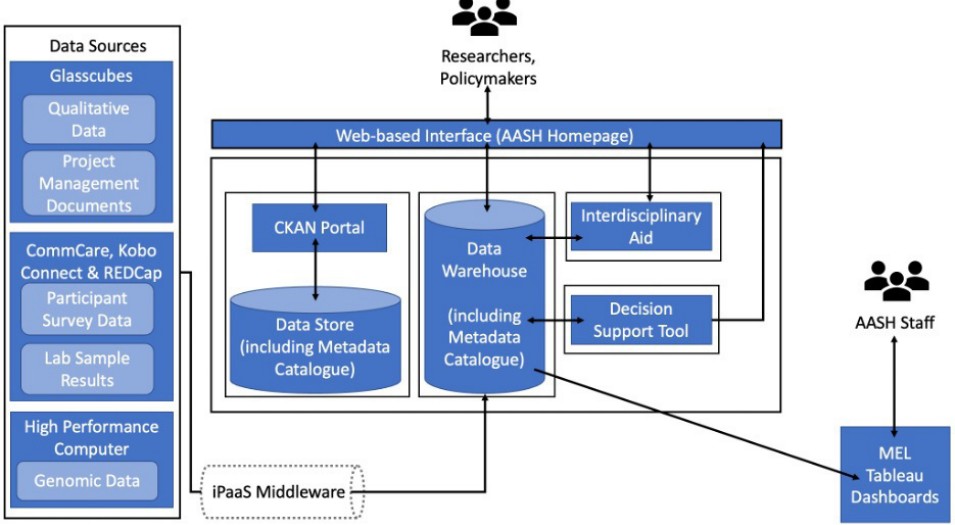

**Figure 2** Visualisation of the All-Hub Data Repository. AASH, Action Against Stunting Hub.

used to organise the different classes of data collected by the AASH into a hierarchy based on their various categories, the types of relationships between them and their potential 'nodes' or areas where disciplines overlap.[12 13]

### The Decision Support Tool

The AHDR aims to provide a decision support tool to enable policy-makers, practitioners and field staff to interrogate AASH data. The tool will predict a child's risk for being on the pathway to stunting. The app will prompt users for a number of input variables which will be sent back to the Python model to predict a traffic light colour. Additionally, bulk-processing functionality will be available via uploading a file of input variables. Users can then download their file, modified with a predicted colour for each row of input variables in the input file. Various visualisations, including geographic visualisation, based on the bulk data, will also be provided.

Figure 2 provides an overview of the interconnected components of the AHDR as outlined above, including the ways in which it integrates inputs held in our project management and communication application (ie, Glasscubes), with our field-data collection software (ie, CommCare, Kobo Connect and REDCAP), and genomic data stored on high-performance computers. As seen below, all data from these sources will be integrated via our iPaaS middleware and sent to our data warehouse. The data warehouse will then feed all structured data into the web-based data portal, where researchers and other key stakeholders can interact with and export AASH data chosen from both thematic and interdisciplinary combinations. Interested parties will ultimately also be able to draw information from the data warehouse using the IA and decision support tool, and our project management staff will have secure access to information related to our project's progress via the MEL Tableau dashboards.

### Patient and public involvement

Participant and public involvement towards the eradication of child stunting is at the core of our project. We are working directly with the local communities in our study countries in India, Indonesia and Senegal, respectively, to understand how stunting and the work of the AASH have impacted their daily lives. Along with this, with a strong focus on user experience and interaction at the core of the AHDR's design, we aim for our work to bring researchers from different disciplines together to share knowledge and expertise, including allowing researchers to upload their analysed datasets for use by others. The AHDR also aims to provide a resource for policy-makers and practitioners to hone interventions that will have the highest possible benefit to the general public.

### ETHICS AND DISSEMINATION

Ethical approval for the AASH, inclusive of the AHDR, was granted by the Ethics Committee of the London School of Hygiene and Tropical Medicine (17915/RR/17513); the Social Science Research Ethical Review Board at the Royal Veterinary College (URN SR2020-0197); and the International Livestock Research Institute Institutional Research Ethics Committee (ILRI-IREC2020-33). Research site-specific approvals were granted by the National Institute of Nutrition (ICMR), Ministry of Health and Family Welfare, Government of India (CR/04/I/2021); Health Research Ethics Committee, University of Indonesia and Cipto Mangunkusumo Hospital (KET-887/UN2.F1/ETIK/PPM.00.02/2019); and the Comité National d'Ethique pour la Recherche en Santé, Senegal (Protocole SEN19/78).

Dissemination will include publications in international peer-reviewed journals; community-level public engagement events, national and international conferences and key stakeholder meetings; and the use of the AHDR by

our vast network of AASH researchers, and ultimately, the wider research and policy-making community. The AASH will seek to make data as open as possible as is consistent with UKRI priorities and Findable, Accessible, Interoperable, Reusable data principles. On publication, study data will be posted in public repositories (preferably domain-specific), or as required by the relevant publication body. Published data will be made available with a preference for minimal licence restrictions (typically CC BY, CC BY-NC, or CC BY-ND) as per UKRI guidance, following assessment for exploitation potential.

**Author affiliations**
[1]Department of Public Health, Environments and Society, London School of Hygiene and Tropical Medicine, London, UK
[2]Department of Population Health, London School of Hygiene and Tropical Medicine, London, UK
[3]School of Computing and Mathematical Sciences, Birkbeck University of London, London, UK
[4]ICMR National Institute of Nutrition, Hyderabad, Telangana, India
[5]Department of Parasitology-Mycology, Cheikh Anta Diop University of Dakar, Dakar, Senegal
[6]Rowett Institute of Nutrition and Health, University of Aberdeen, Aberdeen, UK
[7]London International Development Centre, London, UK

**Acknowledgements** We acknowledge and thank Taylor Downs, Aleksa Krolls, Joseph Sam and Aissatou Diallo at OpenFn who the Action Against Stunting Hub contracted to build the AHDR.

**Contributors** KC-M and CH drafted the manuscript. KC-M, CH, DTT, PW and VK provided critical revision to the manuscript. CH, PW, VK and DTT designed the tool. KC-M, DTT, PW, VK, DY, LFA, MM, AD and FT contributed to the design. All authors read and approved the final manuscript.

**Funding** This work is supported by UKRI-GCRF Action Against Stunting Hub, grant number MR/S01313X/1.

**Competing interests** None declared.

**Patient and public involvement** Patients and/or the public were not involved in the design, or conduct, or reporting, or dissemination plans of this research.

**Patient consent for publication** Not applicable.

**Ethics approval** Ethical approval for the Action Against Stunting Hub, inclusive of the All-Hub Data Repository, was granted by the Ethics Committee of the London School of Hygiene and Tropical Medicine (17915/RR/17513); the Social Science Research Ethical Review Board at the Royal Veterinary College (URN SR2020-0197); and the International Livestock Research Institute Institutional Research Ethics Committee (ILRI-IREC2020-33). Research site-specific approvals were granted by the National Institute of Nutrition (ICMR), Ministry of Health and Family Welfare, Government of India (CR/04/I/2021); Health Research Ethics Committee, University of Indonesia and Cipto Mangunkusumo Hospital (KET-887/UN2.F1/ETIK/PPM.00.02/2019); and the Comité National d'Ethique pour la Recherche en Santé, Senegal (Protocole SEN19/78). Participants gave informed consent to participate in the study before taking part.

**Provenance and peer review** Not commissioned; internally peer reviewed.

**Data availability statement** Data sharing not applicable as no datasets generated and/or analysed for this study. This is a study protocol outlining the development of the Action Against Stunting Hub's data repository, and as such no datasets were generated or used. As part of the outputs of the Action Against Stunting Hub, data will be posted in public repositories, or as required by the relevant publication body. Published data will be made available with a preference for minimal licence restrictions (typically CC BY, CC BY-NC, or CC BY-ND) as per UKRI guidance, following assessment for exploitation potential.

**ORCID iDs**
Kaitlin Conway-Moore http://orcid.org/0000-0003-0128-3922
D M Dinesh Yadav http://orcid.org/0000-0002-8843-3016
Modou Lamin Jobarteh http://orcid.org/0000-0002-7350-6980

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
