## [Reviewer comments · BMJ Paediatrics Open]

ARTICLE DETAILS

TITLE (PROVISIONAL)	
AUTHORS	

VERSION 1 – REVIEW

REVIEWER	Dr. Vandana Prasad Public Health Resource Society, New Delhir H-45 Sector 39 NOIDA Uttar Pradesh 201301 India
REVIEW RETURNED	14-Mar-2024

GENERAL COMMENTS	i would hope that the tool is adequately evaluated as I anticipate many user issues as well as interpretation issues that might arise out of, essentially, an 'auto-mode' analysis of creating summary reports related to risk. in some sense, anthropometry itself is a non-complex observational summary of very complex and child-unique determinants. However, since it will allow researchers a large dataset with many variables, it will certainly help more inter-disciplinary work on nutrition.
---

VERSION 1 – AUTHOR RESPONSE

Thank you for this feedback. We agree that evaluation of the data repository tool will be needed in order to understand how it supports and furthers interdisciplinary research on child stunting. In first making the data repository available to AASH researchers to conduct their analysis, we also hope to be able to work out any major user-related issues.

VERSION 2 – REVIEW

REVIEWER	Dr. Vandana Prasad Public Health Resource Society, New Delhir H-45 Sector 39 NOIDA Uttar Pradesh 201301 India
REVIEW RETURNED	27-Apr-2024

GENERAL COMMENTS	comments have been already made and addressed
---

VERSION 2 – AUTHOR RESPONSE

None